# Acute Kidney Injury by Ischemia/Reperfusion and Extracellular Vesicles

**DOI:** 10.3390/ijms242015312

**Published:** 2023-10-18

**Authors:** Mikkel Ørnfeldt Nørgård, Per Svenningsen

**Affiliations:** Department of Molecular Medicine, University of Southern Denmark, DK-5000 Odense, Denmark; mnoergaard@health.sdu.dk

**Keywords:** ischemia, reperfusion, metabolism, mitochondria, extracellular vesicle release

## Abstract

Acute kidney injury (AKI) is often caused by ischemia-reperfusion injury (IRI). IRI significantly affects kidney metabolism, which elicits pro-inflammatory responses and kidney injury. The ischemia/reperfusion of the kidney is associated with transient high mitochondrial-derived reactive oxygen species (ROS) production rates. Excessive mitochondrial-derived ROS damages cellular components and, together with other pathogenic mechanisms, elicits a range of acute injury mechanisms that impair kidney function. Mitochondrial-derived ROS production also stimulates epithelial cell secretion of extracellular vesicles (EVs) containing RNAs, lipids, and proteins, suggesting that EVs are involved in AKI pathogenesis. This literature review focuses on how EV secretion is stimulated during ischemia/reperfusion and how cell-specific EVs and their molecular cargo may modify the IRI process. Moreover, critical pitfalls in the analysis of kidney epithelial-derived EVs are described. In particular, we will focus on how the release of kidney epithelial EVs is affected during tissue analyses and how this may confound data on cell-to-cell signaling. By increasing awareness of methodological pitfalls in renal EV research, the risk of false negatives can be mitigated. This will improve future EV data interpretation regarding EVs contribution to AKI pathogenesis and their potential as biomarkers or treatments for AKI.

## 1. Introduction

Acute kidney injury (AKI) is a syndrome characterized by an abrupt decrease in kidney function associated with high mortality, impaired organ functions, and the development of chronic kidney disease [1]. The poor patient outcome and a high prevalence of up to 50% in critically ill patients [2] make AKI a global health burden. AKI is clinically classified into pre-renal, post-renal, and intrinsic causes [3], and a significant mechanistic contributor to AKI is ischemia-reperfusion [4,5]. Kidney ischemia, which can happen, e.g., during cardiac surgery or kidney transplantation, severely compromises the kidney’s metabolism, and the reperfusion elicits a robust pro-inflammatory response associated with increased oxidative stress that damages the kidney [6]. Recent studies suggest that the ischemia/reperfusion injury is mitigated by targeted intervention that prevents excessive mitochondrial production of reactive oxygen species (ROS) [7,8]; however, the cellular and molecular mechanisms are unresolved.

Mitochondrial-derived ROS significantly contribute to the cellular secretion of extracellular vesicles (EVs) [9,10]—nanosized vesicles containing proteins, RNA, microRNA (miRNA), and lipids released into body fluids such as blood and urine. EVs comprise a complex heterogeneous vesicle population released from most cell types by several different mechanisms, but technical and experimental challenges make it difficult to define a causal relationship between EVs and AKI progression. Nonetheless, new experimental models and approaches developed within recent years have enabled a deeper insight into EV biology. We will review the current understanding of how ischemia/reperfusion stimulates EV secretion, how EVs contribute to AKI, and highlight some pitfalls associated with kidney EV research uncovered using reporter models.

## 2. Acute Kidney Injury

The kidneys receive up to 25% of the cardiac output, but the oxygen tension in the kidney is generally low [11]. Transient failures of the systematic blood or intra-renal circulation thus increase the risk for detrimental imbalances between oxygen and nutrient delivery to the tubular epithelial cells. Indeed, short-term ischemia and subsequent reperfusion initiate kidney injury. If the damage is mild, repair processes can restore the kidney to normal morphology and function. However, in severe cases, AKI develops into a chronic kidney disease with progressive fibrosis and loss of organ function [12]. The availability of human IRI samples is limited, and most of our knowledge is derived from animal models where transient clamping of the renal pedicles induces IRI. This may confound our IRI knowledge. In contrast to most patients developing AKI, where age and co-morbidities may affect the pathogenesis [13], the IRI models often use healthy young animals. Additionally, experimental kidney injury is usually induced in male animals to circumvent hormonal fluctuations in female mice, which is IRI-protective [14,15]. Nonetheless, there is a significant overlap in the gene transcriptional responses in post-ischemic human and mouse kidneys, indicating a similar acute response to renal IRI [16]. 

### Ischemia-Reperfusion Induces Acute Kidney Injury

The first acute response is characterized by ischemic activation of oxygen-sensitive transcription factors such as hypoxia-induced factor (HIF) and the NF-kB signaling pathway [17,18]. HIF-1α is enzymatically hydroxylated to undergo proteasomal degradation during normoxic conditions, but during hypoxia, this process is inhibited. The hypoxia-induced stabilization of HIF-1α causes its translocation into the cell nucleus, where it binds HIF-1β [19]. The heterodimeric complex binds the genomic hypoxia-responsive elements. It increases transcription of several genes involved in angiogenesis and tissue survival [20] that have protective effects for tissue repair [21]. In addition to HIF signaling, a key regulator of innate and adaptive immune responses, NF-kB is also activated. Renal IRI induces widespread NF-kB activation in renal tubular epithelial and interstitial cells, and genetic inhibition of NF-kB in renal proximal, distal, and collecting duct epithelial cells of mice improved renal function after IRI by attenuating neutrophil and monocyte/macrophage infiltration, which reduced tubular apoptosis and protected renal function [22]. In addition to HIF’s and NF-kB’s impact on the pathogenesis of AKI, ischemia acutely affects mitochondrial function. 

Mitochondria are the primary consumers of molecular oxygen through oxidative phosphorylation. The mitochondrial oxidative phosphorylation creates ATP by reducing oxygen to water, and the electrons for this come from the mitochondrial electron transport chain (ETC). The ETC is localized in the inner mitochondrial membrane and consists of a series of protein complexes and electron carriers that efficiently shuttle electrons from the oxidation of metabolic substrates, e.g., the citric acid cycle, to oxygen. Despite efficient electron transport, a fraction of the electrons prematurely leak from the ETC [23] and contribute to ROS creation. ROS is a term used for the highly unstable and reactive singlet oxygen (^1^O_2_), superoxide anion radicals (O_2_^·−^), hydroxyl radicals (^·^OH), and hydrogen peroxide (H_2_O_2_). While ROS has essential physiological functions, excessive ROS causes oxidative modifications on carbohydrates, lipids, proteins, and DNA, with harmful consequences for cell function and integrity [24], and mitochondrial ROS production is increased during IRI.

The low oxygen availability during kidney ischemia hampers electron flow through the ETC and causes citric acid cycle intermediate accumulation (Figure 1). Succinate levels, in particular, increase significantly during ischemia. The high succinate levels are problematic when the oxygen supply is reestablished by reperfusion in that this reverses the ETC’s electron transport and stimulates excessive mitochondria-derived ROS production [7,8]. Succinate is oxidized to fumarate by succinate dehydrogenase (SDH, Complex II in the ETC), and pharmacological SDH inhibition reduces infarct size in cardiac IRI [8]. Recently, mitochondrial pyruvate dehydrogenase kinase 4 (PDK4) expression was shown to be induced by IRI and genetic and pharmacological PDK4 inhibition before IR, reducing succinate accumulation and kidney damage [7]. Interestingly, the protective effects of PDK4 inhibition were blunted by co-treatment with cell-permeant dimethyl succinate [7]. These observations suggest that the ischemic succinate accumulation is a critical driver for the excessive ROS production during reperfusion that causes kidney injury.

The time course for human kidney IR injury is challenging to establish. Still, transcriptomic studies of human kidney biopsies obtained before ischemia and hours and months after reperfusion also support the critical role of mitochondria [16,25]. While the acute transcriptional program for kidney ischemia and reperfusion was similar between patients, the patients took one of two transcriptional trajectories in the following months: one leading to recovery and one associated with sustained injury [16]. The major difference among the trajectories was that the sustained kidney injury trajectory was associated with mitochondrial dysfunction. Thus, several lines of evidence point to the crucial role of mitochondria in the acute ischemia/reperfusion phase that initiates kidney injury.

The acute ischemia/reperfusion is followed by a rapid inflammatory response initiated by the secretion of pro-inflammatory cytokines (TNF-α, IL-1, IL-6, etc.) and chemokines (MCP, IL-8, etc.) from the kidney’s endothelial and parenchymal cells [26]. The cytokines TNF-α and IL-1 upregulate adhesion molecules like P selectin, intracellular adhesion Molecule 1 (ICAM-1), and vascular cell adhesion molecule 1 (VCAM-1) acutely after IR [27]. The increased expression of adhesion molecules and endothelial cell swelling disrupts the glycocalyx and endothelial monolayer, enhancing inflammatory cell attachment and migration into the injured tissue [28]. These molecular events contribute to the pro-inflammatory response in early AKI after IRI [29,30,31] and enable monocytes/macrophages infiltration within hours [32].

Mouse monocytes are divided into three subsets based on their surface expression of Ly6C. While Ly6C^low^ monocytes mature into M2 macrophages displaying anti-inflammatory properties that contribute to tissue repair, Ly6C^high,^ and Ly6C^int^ are pro-inflammatory monocytes expressing high levels of C-C motif chemokine receptor 2 (CCR2) and C-X3-C motif chemokine receptor 1 (CX3CR1) [33]. Mice deficient in the chemokine receptors CCR2 [34] or CX3CR1 [34,35] are protected from AKI since the egress from bone marrow and subsequent kidney infiltration of inflammatory Ly6C^high^ monocytes are inhibited. Ly6C^high^ monocyte infiltration into the kidney after IR is also reduced in mice deficient in the complement component receptors C3a or C5a [36]. The similar AKI protection between the different knockout mice indicates that complement factors and chemokine signals converge on Ly6C^high^ monocytes and coordinate their function. Although it has not been directly determined how EVs contribute to kidney IRI pathogenesis, several lines of evidence indicate that EV release could be stimulated by ischemia/reperfusion, and be involved in an immediate response, and, for example, provide guidance cues to pro-inflammatory monocytes.

## 3. Extracellular Vesicles in Acute Kidney Injury

The kidney’s tubular epithelial cells are critical for several functions, and EVs secreted from their apical and basolateral membrane compartments contribute to the circulating plasma and urine pools of EVs [37,38]. Although the fraction of plasma and urine EVs (uEVs) derived from the kidney has not been directly determined, bulk RNA sequencing of EV-associated RNAs has enabled an estimation of the cell-specific EV abundances in different biofluids [39,40,41,42,43,44]. For example, the exoRbase database has used the abundance of kidney tubular epithelial-specific markers to estimate that ~1 in every 10,000 plasma EVs is kidney epithelial-derived [39,42]. With (62 ± 17) × 10^12^ plasma EVs in humans, this yields ~10^9^ kidney tubular-derived EVs or ~1 µg kidney-derived EVs per L plasma [9]. Similar approaches tourine EVs have shown that bladder [43] and proximal tubule cells [44] contribute most to the uEV pool. While this approach has not been verified for accuracy, the urine data are consistent with proteomic analyses showing that most proteins identified in uEVs are expressed in the urogenital system [45]. Moreover, Blijdorp et al. directly compare the uEV proteome of urine sampled normally or through a nephrostomy drain [46]. This elegant approach showed that of the 2814 identified proteins, only twelve were not detected in the nephrostomy drain samples [46], suggesting that most uEVs are present in the urine after the exit of the kidney.

It has been technically challenging to determine whether plasma EVs enter the urine. While early reports showed urine excretion of blood-administered EVs in anesthetized mice and rats [47,48], newer methods to study endogenous EVs under physiological conditions in mice do not fully support this conclusion. We have created an EV reporter mouse in which EVs derived from Cre recombinase-expressing cells are labeled with the truncated EV marker CD9 fused to an enhanced green fluorescent protein (EGFP) [49]. The extra-vesicular EGFP enables affinity enrichment of cell-specific EVs obtained under physiological conditions from different biological fluids, such as plasma and urine. Using the EV reporter mouse crossed with Cre expression in all cell types (CMV-Cre), cardiomyocytes (αMHC-MerCreMer), or kidney epithelium (Pax8-Cre), we demonstrated that cardiomyocyte-derived EVs were readily detectable in plasma samples but not in urine. On the other hand, kidney epithelial-derived EVs are abundantly present in urine but not in plasma [49]. This observation does not exclude that some plasma EVs may enter the urine; however, it suggests that the glomerular filtration barrier establishes a charge- and size-selective filter, hampering free EV filtration. Thus, the kidney epithelium is the major contributor to the uEV proteome.

### 3.1. EV Secretion during Ischemia/Reperfusion

All kidney tubular segments contribute to the uEV pool, and EVs primarily enter the urine by apical secretion [38], but it is still unknown how the uEV secretion rate is controlled in health and disease. 

Hypoxia increases EV release in vitro [50,51,52], but the mechanisms are not entirely understood. The HIF-1 dependence appears to involve Rab GTPases. The Rab GTPases are essential for membrane budding and fusion with the plasma membrane. HIF-1a activation increased Rab27a mRNA expression in B cells [53], and in breast cancer cells exposed to 24 h hypoxia, and HIF-1 induces small Rab GTPase RAB22A expression and augments EV release [54]. Furthermore, it has been suggested that increased levels of Rab27a can promote membrane fusion, while decreased Rab7 can stimulate the fusion of MVBs with lysosomes, leading to the degradation of EVs under hypoxic conditions [55]. However, expression of oxygen-insensitive HIF-1α or pharmacological intervention with HIF-1α stabilizer during normoxia is insufficient to increase EV release [10,51]. Like most EV research, the relation between hypoxia and EV release is mainly investigated in vitro, and future research is needed to understand HIF regulation of EV secretion. Additionally, hypoxia triggers HIF-independent signaling pathways such as STAT3 [56], NF-kB [57], and mTOR [58]. mTOR activation, however, decreases EV secretion by affecting the subcellular distribution of the endolysmal system [59]. Nonetheless, recent findings indicate that mitochondrial-derived ROS is important for stimulation of EV secretion [10,60] (Figure 1).

In cultured kidney epithelial cells, we have demonstrated the pharmacological interventions that increase the electron flow through the mitochondrial ETC and augment EV secretion by a ROS-dependent mechanism [10]. Additionally, the estimated cell type-specific EV secretion rates of human blood cells are strongly correlated with their respiratory capacity, as determined by the activity of the protein complexes in the ETC [9]. These data suggest that the mitochondrial ETC’s activity and ROS production could be potent stimulators for EV secretion, but whether this form of EV stimulation contributes to basolateral and apical EV secretion from kidney epithelial cells is not known. 

Interestingly, the uEV and creatinine concentrations are closely correlated in urine samples from healthy humans [46,61]. As stated above, EVs are not freely glomerularly filtered and are derived mainly from cells lining the urogenital system [45]. In contrast to plasma EVs, creatinine is freely filtered at the glomerulus, and creatinine is used to estimate the glomerular filtration rate (GFR). The correlation between urine creatinine and uEVs thus implies that kidney epithelial EV secretion is linked with the glomerular filtered load. Increased GFR causes enhanced tubule salt transport, which is energy-demanding and increases oxygen consumption. However, whether the kidney epithelial cells’ mitochondrial ETC activity and ROS production cause the correlation between creatinine and uEV concentration remains to be determined. Nonetheless, mitochondrial-derived ROS production is augmented during ischemia and early reperfusion [7,8]; thus, acutely, the cellular EV secretion may be augmented by ischemia/reperfusion as a cell protective mechanism or as a form of intercellular communication that contributes to the development of AKI. 

### 3.2. What Are the Functions of the Secreted EVs?

It is still unknown what the exact functions of the secreted EVs are and whether EVs have harmful or protective roles, e.g., by functioning as cellular waste managers or intercellular signaling entities (Figure 2). A critical parameter for the EVs role as intercellular signaling pathways, irrespective of whether they are taken up by recipient cells or act as receptor ligands, is their biodistribution. The circulatory time for endogenous EVs is not known. Still, a systematic review of pharmacokinetic studies of labeled EV in rats and mice suggests a mean plasma residence time of 30–50 min [62]. In humans, indirect measures of plasma half-life corroborate this finding [63,64], suggesting that secreted EVs “survive” several passages through the circulatory system. The EV circulatory half-life is significantly increased by chemical chlodronate liposome treatment in mice, suggesting that macrophages are the dominant recipient cell type [65]. This could potentially mean that EVs have important modulatory functions in the immune system. 

In the AKI inflammatory process, EVs may provide guidance cues to inflammatory cells and play an important role in the rapid infiltration of monocytes into the kidney after IRI. This EV signaling mechanism has, for example, been demonstrated in primary human lymphatic endothelial cells after exposure to the inflammatory cytokine TNF-α [66]. Here, the endothelial cells release inflammatory EVs that can induce the formation of dynamic exploratory cell protrusions via membrane-bound CX3CL1 in a chemokinetic G protein-coupled receptor signaling-dependent manner in recipient human dendritic cells [66]. Similar mechanisms could be activated in monocytes due to their expression of CX3CR1 and provide guidance cues together with inflammatory effector molecules such as C3a and C5a in the kidney after IRI. Thus, EVs may be important signaling molecules for the development of AKI by affecting the infiltration of monocytes. 

In vivo, the demonstration of EV cell-to-cell communication in the kidney is limited. Still, in vitro renal tubular epithelial cell-derived EVs can transfer EVs from proximal tubular cells to distal tubular and collecting ducts cells [67], between cultured murine kidney collecting ducts (mCCDc11) cells [68], and between human bone marrow mesenchymal stem cells and cultured tubular epithelial cells, where they stimulate proliferation [69]. Although the EV concentration used in in vitro studies tends to be higher than the physiological concentration, a more physiologically relevant setup using a Boyden chamber supports intercellular EV transfer between proximal tubular cells and fibroblasts [70]. The EVs secreted from hypoxic proximal tubular cells induced fibroblast proliferation, TGF-β1 expression, α-smooth muscle actin (α-SMA expression), F-actin expression, and type I collagen (α1-chain) production [70], which supports the idea of EVs involved in cell–cell communication. Nonetheless, physiological in vivo transfer between cells in the kidney has still not been shown.

Another function of the secreted EVs could be cellular waste management. Consistent with this, inhibition of cellular EV secretion impairs health [71]. Using a paired analysis of human kidneys and uEVs, we found no correlation between segment-specific markers in kidney tissue and uEVs [72]. The variability in uEV protein abundances was significantly higher than the kidney abundances [72]. The reason for this is likely multifactorial; however, one interpretation of this observation is that protein abundance in kidney tissue is under homeostatic control, and the uEV serves as a cell-protective mechanism. 

The secreted EVs may play an important regulative role in the relationship between oxidative stress and AKI development. On the one hand, EVs contain antioxidants [73,74,75] that may confer cell protection to recipient cells. On the other hand, EVs can transfer pro-oxidants, aggravating the injury response [76]. Additionally, EVs can affect the rigidity of the cell membrane, resulting in decreased ROS-producing abilities of recipient cells, such as observed for neutrophils and monocytes [77,78]. Thus, EVs may indirectly regulate the cellular response to AKI by affecting ROS production in immune cells. Several roles of kidney-derived EVs have been suggested for the development of AKI; however, there is a lack of tools that can be used to directly intervene with EV secretion to gain more mechanistic insight into the role of EVs.

## 4. Pitfalls in Kidney EV Research—EV Loss during Tissue Preparation

Unwrapping the biological roles of kidney-derived EVs is challenging since EVs are released from most cell types. This hampers our ability to dissect cell-specific EV release and its target cells in vivo. EVs are commonly tracked with organic vital dyes, such as PKH2 and PKH26, providing excellent spatial resolution [79,80,81]. The staining procedure can be performed on parental cells or directly on isolated EVs, followed by washing procedures to remove unbound dye. However, this can lead to unspecific binding and the formation of micelles with similar size and density as EVs, owing to their lipophilic nature [82]. Lipid-anchored fluorophores also label lipoproteins, abundantly present in plasma [83]. To overcome this, we and other researchers have used genetic labeling of EV proteins fused with a fluorescent protein such as red fluorescent protein (RFP), GFP, or GFP pH-sensitive derivates like pHluorin [49,84,85,86]. Genetic labeling overcomes some limitations of the artificial fluorescence EV signal associated with classical EV labeling using fluorescent dyes. It allows cell-specific EV labeling, but the fluorescent intra- or extravesicular tags may modify EV function and biodistribution by affecting normal cell–cell signaling functions of EV proteins. Nonetheless, the use of our EV reporter mouse has also revealed an additional pitfall.

As we previously reported [49], kidney epithelial-derived CD9truc-EGFP EV signals are significantly affected by the preparation of frozen kidney sections before fluorescent microscopy, leaving only minimal signal accumulated in the glomerulus with no genetic EGFP expression [49]. This was specific for the kidney epithelium in that CD9truc-EGFP was abundantly expressed in frozen sections of hearts from cardiomyocyte-specific Cre mice [49]. Preparing tissue sections for cryosectioning involves buffers with high osmolarity, which may cause severe cell shrinkage in epithelial cells due to their high water permeability. Consistent with this notion, the CD9truc-EGFP loss was mitigated in paraffin-embedded kidney sections [49]. Others have also observed the release of EVs during tissue preparation, and the human vitreous humor-derived EV signal was lost during formalin fixation due to temperature-sensitive crosslink reversal. Nonetheless, the EV signal was retained by fixation with 1-ethyl-3-(3-dimethyl aminopropyl) carbodiimide (EDC) [87] and suggests that we need more knowledge on optimal fixation of EVs to quantify tissue abundance and cell-specificity of EVs reliably. This is a critical methodological pitfall to keep in mind in that significant transfer of kidney-derived EVs may occur during tissue preparation, which can confound our interpretation of EV biology.

## 5. Conclusions and Perspectives

The exact role of EVs in AKI pathogenesis is still unknown, but evidence from several in vitro and in vivo models indicates EVs could play many roles. Although several stimuli may affect EV release during AKI, the release of EVs may be potently stimulated by ischemia/reperfusion. These EVs could have essential functions, such as the recruitment of pro-inflammatory monocytes, the transfer of antioxidants, or ROS, and waste management. Moreover, EVs are a heterogeneous population of vesicles, and some EV subpopulations may mediate cell-protective mechanisms while others are harmful. The development of new experimental tools to study the different EV populations in vivo will be important to elucidate which EVs are harmful, and which are protective.

The potential mechanistic involvement of EVs in AKI pathogenesis, combined with the molecular cargo and information on their cell of origin, also qualifies EVs as promising candidates for biomarkers and therapeutic approaches concerning AKI. Despite their low abundance, kidney epithelial cell-derived EVs can be isolated from plasma, and the uEVs are primarily derived from cells lining the urogenital systems. Thus, kidney-derived EVs and uEVs have great potential as non-invasive biomarkers to monitor AKI. EVs are also interesting for potential AKI treatments, and the possibility to engineer the EVs hold great promise for effective EV treatment in the future.

Thus, future improvements in EV research will lead to a more accurate understanding of AKI pathogenesis and provide a stronger foundation for AKI biomarkers and treatments for more efficient kidney protection.

## Figures and Tables

**Figure 1 ijms-24-15312-f001:**
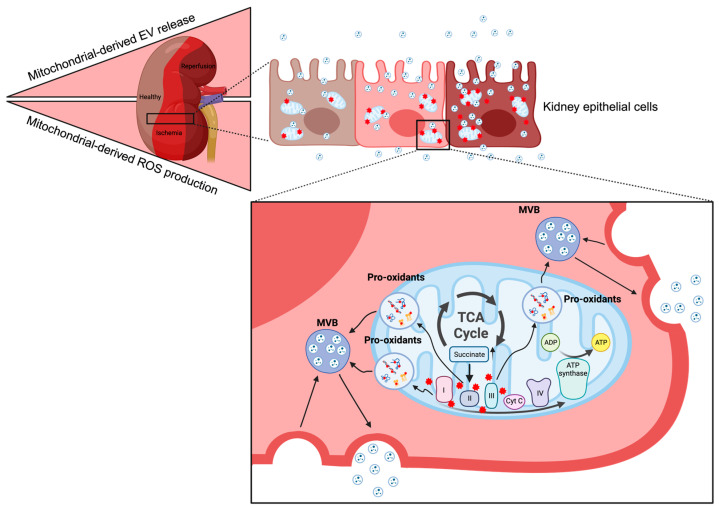
Ischemia/reperfusion causes increased mitochondrial-derived ROS (red stars) production in kidney epithelial cells due to accumulation of specific citric acid cycle (TCA) intermediates, such as succinate. Increased ROS oxidizes proteins, lipids, and DNA and potentially stimulates EV secretion (Created with BioRender.com).

**Figure 2 ijms-24-15312-f002:**
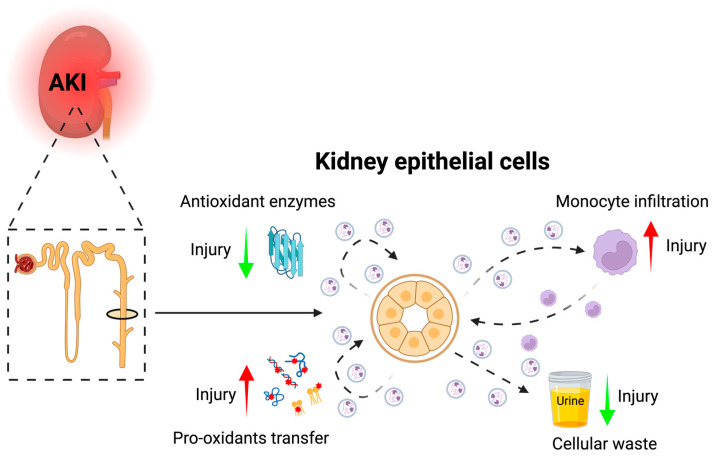
Kidney epithelial-derived EVs could potentially be involved in AKI development through cell-to-cell transfer that aggravates (red arrows) or inhibits (green arrows) ROS induced injury in recipient cells through transfer of pro- and anti-oxidants, as mediators of monocyte recruitment, and as an important injury handling mechanism by release of damaged cellular content (Created with BioRender.com).

## Data Availability

Data sharing not applicable.

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
