# Peer review of "Acute Kidney Injury by Ischemia/Reperfusion and Extracellular Vesicles"

_ijms, 2023, doi:10.3390/ijms242015312_

Round 1

Reviewer 1 Report

Acute kidney injury is a wide term, ischemia-reperfusion injury is only one of the causes.

The review focuses on IRI and the title should reflect this.

In the abstract, but also in the manuscript, ROS are mentioned, but these are surely not the major pathomechanism of AKI and vesicles can be formed without ROS as well.

the text should focus on the vesicles in IRI, everything else should be omitted. 

ROS are mostly very, very short-living. It is very unlikely that vesicles "transfer ROS" as indicated in figure 2.

The authors need to prepare further explanatory figures related directly to the topic - especially, the origin of vesicles and their transport should be explained.

It is of utmost importance whether vesicles in the urine are linked to the function or to the kidney damage.

The most important part of the review is the chapter Pitfalls.

"Together, this indicates that EVs and uEVs have great potential as non-invasive biomarkers. " Read the preceding sentence. It does not indicate that the vesicles would be a good biomarker...

The conclusion and perspective are full of results that should not be there. And the perspective - the suggestions for future research are missing.

Author Response

Comment 1:

Acute kidney injury is a wide term, ischemia-reperfusion injury is only one of the causes.

The review focuses on IRI and the title should reflect this.

Response 1:

Thank you for your efforts in reading and commenting on our manuscript. We agree with your comment on the title, and has therefore changed the title to:

Acute kidney injury by ischemia/reperfusion and extracellular vesicles

Comment 2:

In the abstract, but also in the manuscript, ROS are mentioned, but these are surely not the major pathomechanism of AKI and vesicles can be formed without ROS as well.

Response 2

Thank you for the comment. We have added the following to the abstract:

Line 9: transient

Line 11: ,and together with other pathogenic mechanisms.

Line 41-42: by several different mechanisms

Comment 3:

the text should focus on the vesicles in IRI, everything else should be omitted. 

Response 3:

We believe that if we omit everything else there is a risk for us to limit the number of readers since only EV experts with knowledge on AKI (or vice versa) would benefit from it. We have therefore given a broad introduction to kidney IRI to increase the readability for a larger audience.

Comment 4:

ROS are mostly very, very short-living. It is very unlikely that vesicles "transfer ROS" as indicated in figure 2.

Response 4:

Thank you for the comment. We agree that this could be misleading, and we have changed it to “transfer of pro-oxidants” in the figures and text (line 293).

Comment 5:

The authors need to prepare further explanatory figures related directly to the topic - especially, the origin of vesicles and their transport should be explained.

Response 5:

We have added new information to Figures 1 and 2. We focus on the initial IR mechanisms, and here mouse studies (see ref 7 and 22) indicate that the epithelial cells play a significant role. We have drawn the tubular epithelial cells as the EV senders to highlight this.

Comment 6:

It is of utmost importance whether vesicles in the urine are linked to the function or to the kidney damage.

Response 6:

We are unsure what this comment means, and we believe that EVs are linked to both function and damage. As we still do not have the experimental tools to decipher which EV subtype is doing what this is still an open question. We have, however, added to the conclusion and perspectives (line 341-343): Development of new experimental tools to study the different EV populations in vivo will be important to elucidate which EVs are harmful and which are protective.

Comment 7:

The most important part of the review is the chapter Pitfalls.

Response 7:

Thank you.

Comment 8:

"Together, this indicates that EVs and uEVs have great potential as non-invasive biomarkers. " Read the preceding sentence. It does not indicate that the vesicles would be a good biomarker...

Response 8:

We are not sure what the “preceding sentence” refers to; however, we believe you are referring to the statement in the manuscript about DM with and without proteinuria. In that study, no difference in cell-specific EV abundance was detected, so differences in the EV’s molecular cargo will likely arise from changes in expression level and not EV secretion rate. Nonetheless, to accommodate our following comment, we have shortened the conclusion and perspectives section, and removed this statement.

Comment 9:

The conclusion and perspective are full of results that should not be there. And the perspective - the suggestions for future research are missing.

Response 9:

Thank you. We have revised the two small paragraphs on EV biomarker and treatment and removed the references.

Reviewer 2 Report

This manuscript, entitled “Acute kidney injury and extracellular vesicles: cell-to-cell communication, biomarkers, and pitfalls”, reviewed the relationship between Acute kidney injury and EV, Obviously, the authors put a lot of effort into this manuscript, However, there some major points needed to be addressed before publication:

1, The demonstration of EV in intercellular communication in the kidney in the review is limited

2, The review did not explore the role of EV as a biomarker.

3, I think most of the researchers who study EV know that EV loss during tissue preparation, why is it a pitfall?

Author Response

This manuscript, entitled “Acute kidney injury and extracellular vesicles: cell-to-cell communication, biomarkers, and pitfalls”, reviewed the relationship between Acute kidney injury and EV, Obviously, the authors put a lot of effort into this manuscript, However, there some major points needed to be addressed before publication:

Response: 

Thank you for your time and comments on our manuscript.

1, The demonstration of EV in intercellular communication in the kidney in the review is limited

Response:

Yes, we agree but most research in the intercellular communication has been done in vitro or by infusion of isolated EVs. We are unsure how well this recapitulates the in vivo situation, and we have not been able to identify studies where endogenously produced EVs are tracked. We have, therefore, highlighted potential roles of EVs based on the available data.

2, The review did not explore the role of EV as a biomarker.

Response:

Thank you for the comments. We have changed the title of the review (see commentary to Reviewer 1) and removed the word biomarker. We have shortened the paragraph on biomarkers in the conclusion and perspectives section.

3, I think most of the researchers who study EV know that EV loss during tissue preparation, why is it a pitfall?

Response:

You are probably correct; however, AKI and the remaining EVs researchers may not be aware of this issue. We think this is a critical pitfall to highlight in that the EVs released during tissue preparation could be taken up by other cells and confound our knowledge on intercellular communication. We have added the following to the manuscript (line 331-333): 

This is a critical methodological pitfall to keep in mind in that significant transfer of kidney-derived EVs may occur during tissue preparation, which can confound our interpretation of EV biology.

Round 2

Reviewer 2 Report

Agree to publish